## SHORT REPORT

# Correcting gradient-based interpretations of deep neural networks for genomics

Antonio Majdandzic[1], Chandana Rajesh[1] and Peter K. Koo[1*]

*Correspondence:
koo@cshl.edu

[1] Simons Center for Quantitative
Biology, Cold Spring Harbor
Laboratory, 1 Bungtown Road,
Cold Spring Harbor, NY, USA

**Abstract**

Post hoc attribution methods can provide insights into the learned patterns from deep neural networks (DNNs) trained on high-throughput functional genomics data. However, in practice, their resultant attribution maps can be challenging to interpret due to spurious importance scores for seemingly arbitrary nucleotides. Here, we identify a previously overlooked attribution noise source that arises from how DNNs handle one-hot encoded DNA. We demonstrate this noise is pervasive across various genomic DNNs and introduce a statistical correction that effectively reduces it, leading to more reliable attribution maps. Our approach represents a promising step towards gaining meaningful insights from DNNs in regulatory genomics.

**Keywords:** Deep learning, Regulatory genomics, Model interpretability, Attribution methods, Explainable AI

## Background

Deep neural networks (DNNs) have demonstrated impressive performance across a wide variety of sequence-based prediction tasks in genomics, taking DNA sequences as input and predicting experimentally measured regulatory functions [1–3]. To gain insights into the features learned by DNNs, post-hoc attribution methods provide an importance score for each nucleotide in a given sequence; they often reveal biologically meaningful patterns, such as transcription factor binding motifs that are essential for gene regulation [4, 5]. Attribution methods also provide a natural way of quantifying the effect size of single-nucleotide mutations, both observed and counterfactual, which can help to prioritize disease-associated variants [6, 7].

 Some of the most popular attribution methods are gradient-based, where partial derivatives of the output with respect to the inputs are used, including saliency maps [8], integrated gradients [9], SmoothGrad [10], and expected gradients [11]. However, in practice, attribution methods often produce noisy feature importance maps with spurious importance scores [12, 13]. This makes it difficult to deduce hypotheses of which patterns drive model predictions, which can then be validated with carefully designed

*in silico* experiments [5, 14]. Many factors that influence the efficacy of attribution maps have been identified empirically, such as the smoothness properties of the learned function [10, 15, 16] and learning (non-)robust features [17–19]. However, the origins of all noise sources that afflict attribution maps are not yet fully understood.

Here, we identify a previously overlooked source of noise in input gradients when the input features are categorical variables. Then, we introduce a simple, but effective, statistical correction and demonstrate that it improves attribution-based explanations across various DNNs that span a wide range of prediction tasks in regulatory genomics.

## Results and discussions

### Off-simplex gradients introduce random noise

Even though DNNs can learn a function everywhere in Euclidean space, one-hot encoded DNA is a categorical variable that lives on a lower-dimensional simplex. A DNN can learn a meaningful predictive function near the data support, i.e., on the simplex, but it has freedom to express any arbitrary function behavior off the simplex where no data exists. Since held-out test data also lives on this simplex, DNNs can still maintain good generalization performance. However, random off-simplex function behavior can introduce a random gradient component orthogonal to the simplex, which manifest as spurious noise in the input gradients (Fig. 1a). This, in turn, can make it more challenging to interpret learned motif patterns or trust variant effect predictions from gradient-based attribution analysis.

To minimize the impact of off-simplex gradient noise, we introduce a simple statistical correction based on removal of the random orthogonal gradient component. For a one-hot sequence, $\mathbf{x} \in \{A\}^L$, with $A$ categories (e.g., 4 for DNA) and length $L$, the gradient ($\mathbf{G} \in \mathbb{R}^{L \times A}$) of the model's prediction with respect to the $l$th position along the sequence and nucleotide index $a$ can be corrected according to: $G_{l,a}^{\text{corrected}} = G_{l,a} - \mu_l$, where $\mu_l = \frac{1}{A} \sum_a G_{l,a}$ (see the "Methods" section for derivation). This proposed gradient correction—subtracting the original gradient components by the mean gradients across components for each position—is general for all data with categorical inputs, including DNA, RNA, and protein sequences.

### Gradient correction improves attribution maps quantitatively

To demonstrate the efficacy of our proposed gradient correction, we systematically evaluated attribution maps before and after the correction for various convolutional neural networks (CNNs) trained on synthetic genomics data that recapitulates a billboard model of gene regulation (see the "Methods" section). We also performed a qualitative evaluation of the gradient correction on various CNNs trained on prominent types of regulatory genomic prediction tasks, including single-task and multi-task binary classification and quantitative regression at various resolutions, using data from a diverse set of high-throughput functional genomics assays measured in vivo.

First, using synthetic data, which provides base-resolution knowledge of ground truth motif patterns, we quantitatively assessed the efficacy of various attribution maps, including saliency maps, integrated gradients, SmoothGrad, and expected gradients (see the "Methods" section). Strikingly, we found that the gradient correction consistently yields a substantial improvement in the quality of attribution maps across various

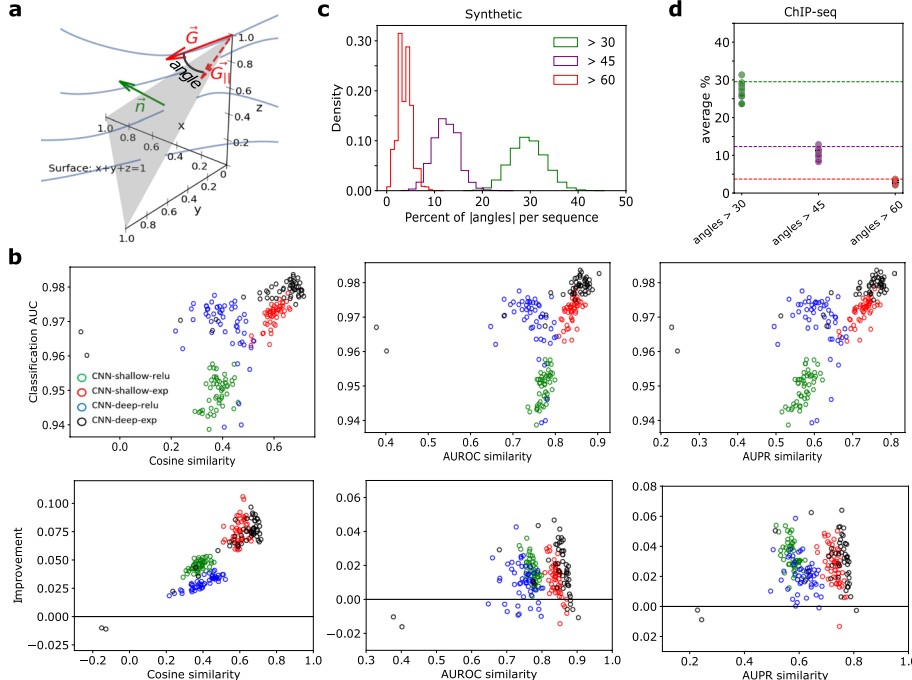

**Fig. 1** Gradient correction performance. **a** Toy diagram of geometric relationship between the input gradient and the simplex defined for 3-dimensional categorical data. Blue curves represent gradient lines of a hypothetical learned function. Gray plane represents the data simplex. The red vector represents the gradient pointing off of the simplex. **b** Performance comparison on synthetic data. (Top row) Scatter plot of interpretability performance measured by different similarity scores versus the classification performance (AUC) for saliency maps. (Bottom row) Interpretability improvement for saliency maps for different similarity metrics when using gradient correction. Improvement represents the change in similarity score after the gradient correction. Each point represents 1 of 50 trials with a different random initialization for each model. **c** Histogram of the percentage of positions in a sequence with a gradient angle larger than various thresholds for a deep CNN with ReLU activations (CNN-deep-relu) trained on synthetic data. **d** Scatter plot of the percentage of positions in a sequence with a gradient angle larger than various thresholds for CNN-deep-relu trained on ChIP-seq data. Each point represents the average percentage across all test sequences for each ChIP-seq dataset. For comparison, horizontal dashed lines indicate the mean value from the corresponding analysis using synthetic data in **c**

similarity metrics (see Fig. 1b for saliency maps and Additional file 1: Fig. S1 for other attribution methods).

### Larger off-simplex angles are associated with spurious noise

Next, we visualized the density of angles between the gradient and the simplex, which highlights the extent of the off-simplex random noise (see the "Methods" section). We found that the distribution of angles is mostly zero-centered but their width varies from model to model (Additional file 1: Fig. S2). Even with the enormous freedom to express arbitrary functions off the simplex, the function that is often learned largely aligns with the simplex.

By focusing on large angles, we found that each attribution map contains about 5–15% of positions with a gradient angle larger than 60°; about 10–20% of positions have angles greater than 45°; and about 20–40% of positions have angles greater than 30° (Fig. 1c, Additional file 1: Fig. S3). A similar observation was made for various

CNNs trained on ChIP-seq data (Fig. 1d and Additional file 1: Figs. S4). This suggests that large angles between the gradients and the simplex are pervasive in attribution maps.

To assess how well the gradient correction works as intended, we compared the improvement of the attribution maps upon correction as a function of the angle magnitude (see the "Methods" section). Across both, CNNs trained on either synthetic data or ChIP-seq data, we found that positions with larger angles indeed yield improved attribution scores in true-positive positions, while background positions led to a decrease in spurious attribution scores (Additional file 1: Figs. S2 and S5).

### Gradient correction generates qualitatively more interpretable attribution maps

To assess whether gradient-corrected attribution maps better align with our notion of intepretability, we also performed a visual comparison. Interestingly, positions within and directly flanking the motif patterns exhibit a high degree of spurious noise in attribution maps without the gradient correction. Many of these seemingly "spurious" nucleotides are associated with gradients that exhibit large angular deviations from the simplex (Fig. 2a, Additional file 1: Fig. S6). Upon correction, attribution maps tend to visually yield cleaner motif definition.

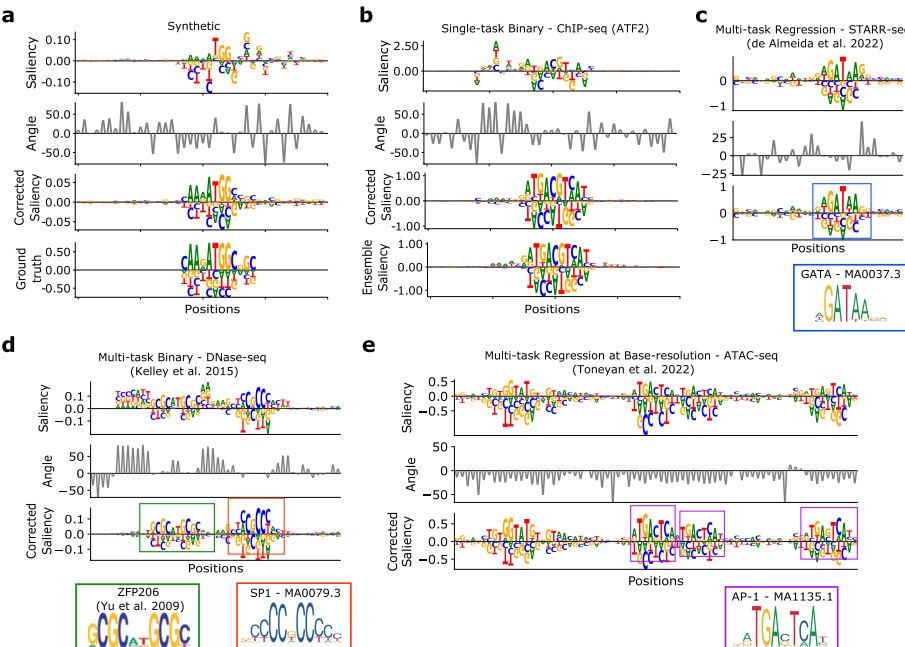

**Fig. 2** Visualizing the gradient correction. Sequence logo of the uncorrected saliency map (top row), gradient angles at each position (second row), and corrected saliency map (third row) for a patch from representative test sequences. **a**, **b** CNN-deep-relu trained to make binary predictions on **a** synthetic data and **b** ChIP-seq data for ATF2 protein in GM12878. The sequence logo of ground truth is shown for CNN-deep-exp for **a** synthetic data. **b** An ensemble average saliency map is shown in lieu of ground truth (bottom row). **c**–**e** A similar plot is made for a **c** DeepSTARR model trained to predict enhancer activity via STARR-seq data, **d** Basset model trained to make binary predictions of chromatin accessibility sites via DNase-seq data, and **e** CNN model trained to predict base-resolution read-coverage values from ATAC-seq data in PC-3 cell line. **c**–**e** A colored box and a corresponding sequence logo of a known motif from JASPAR [20] (with a corresponding ID) or Ref. [21] are shown for comparison

This phenomenon was also observed across other CNNs trained on various high-throughput functional assays measured in vivo, including CNNs trained to predict transcription factor ChIP-seq peaks as a single-task binary classification (Fig. 2b, Additional file 1: Fig. S7), a DeepSTARR model trained to predict quantitative levels of enhancer activity measured via STARR-seq [5] (Fig. 2c, Additional file 1: Fig. S8), and a Basset model trained to predict chromatin accessibility sites across 161 cell-types/tissues as a multi-task binary classification [22] (Fig. 2d, Additional file 1: Fig. S9).

We also found that gradient correction worked well for various base-resolution CNNs trained to predict quantitative levels of normalized read-coverage of 15 ATAC-seq datasets [23] (Fig. 2e, Additional file 1: Fig. S10). Interestingly, the attribution maps of CNNs trained to predict read-coverage down-sampled at 32-bin resolution, on average, exhibited more noticeable improvements with the gradient correction compared to base-resolution CNNs. The initial attribution maps (before the correction) better captured known motifs for the base-resolution CNNs, especially when exponential activations was employed in the first layer [19] (Additional file 1: Fig. S10). Strikingly, the extent of off-simplex gradient angles, and hence the occurrence of random attribution noise, was observed consistently across all of the models and datasets that were investigated (Additional file 1: Figs. S7-S10).

### Additional observations

Upon further investigation, we found that the magnitude of the random initialization plays a major role, with larger random values increasing the extent of off-simplex gradients for both DNNs trained on synthetic and real data (Additional file 1: Note S1). In addition, the gradient correction can be utilized as a regularizer to guide function behavior to align with the simplex during training (Additional file 1: Note S2).

### Conclusions

Attribution methods can provide insights into the *cis*-regulatory syntax learned by genomic DNNs and help to prioritize disease-associated variants. However, unregulated off-simplex function behavior, which arises due to how DNNs fit one-hot DNA sequences, introduces noise in gradient-based attribution maps, which obfuscates biological signals from spurious noise. Our proposed gradient correction is an effective solution to address this issue and it is simple to implement with a single line of code.

Our proposed gradient correction is an effective statistical correction. However, in individual cases, corrections can be subtle, even when a large off-simplex gradient is observed. In addition, many large angles can be associated with positions that have low attribution scores, and thus may not result in noticeable changes. Interestingly, we observed that the largest corrections occur when the attribution scores at a given position are either all positive or all negative (Additional file 1: Figs. S8-S10). In such cases, the gradient correction centers and reduces the attribution scores. On the other hand, attribution methods that are based only on forward propagation, such as *in silico* mutagenesis, do not require this corrections as the DNN's behavior off the simplex does not affect predictions; all data, including test data, lives on the simplex. Moreover, while we demonstrate the gradient correction

on DNNs trained with DNA sequences, it should extend to other data types based on categorical input variables, such as protein and RNA sequences.

Moving forward, it would be beneficial to explore training strategies that can directly address the off-simplex function behavior of DNNs, such as *mixup* [24], *manifold mixup* [25], and *randomized smoothing* [26].

Importantly, the gradient correction only addresses noise associated with erratic function behavior off the simplex. This correction is not a "magic bullet"; it cannot correct other noise sources that afflict attribution analysis. Throughout this study, we rarely observed attribution maps appearing visually worse after the gradient correction. Hence, we recommend that it should always be applied for its benefit in improving the reliability of gradient-based attribution analysis.

## Methods

### Gradient correction for DNA sequences—derivation

Let us consider DNA sequences as inputs to DNNs, which are represented as one-hot encoded arrays of size $L \times 4$, having 4 nucleotide variants (i.e., {A, C, G, T}) at each position of a sequence of length $L$. One-hot encoded data naturally lends itself to a probabilistic interpretation, where each position corresponds to the probability of 4 nucleotides for DNA or 20 amino acids for proteins. While the values here represent definite/binary values, these one-hot representations can also be relaxed to represent real numbers—this is a standard view for probabilistic modeling of biological sequences, where the real numbers represent statistical quantities like nucleotide frequencies. Each position is described by a vector of 4 real numbers, given by $x, y, z, w$. The probability axiom imposes that each variable is bound between 0 and 1 and their sum is constrained to equal 1, that is

$$x + y + z + w = 1. \tag{1}$$

This restricts the data to a simplex of allowed combinations of $(x, y, z, w)$, and Eq. 1—being an equation of a 3-dimensional (3D) plane in a 4D space—defines this simplex. Importantly, an issue arises with input gradients from how DNNs process this data.

The input gradients can be decomposed into two components: the component locally parallel to the simplex, which is supported by data, and the component locally orthogonal to this simplex, which we surmise is unreliable as the function behavior off of the simplex is not supported by any data. Thus, we conjecture that removing the unreliable orthogonal component from the gradient via a directional derivative, leaving only the parallel component that is supported by data, will yield more reliable input gradients. Without loss of generality, we now illustrate this procedure and derive a formula for this gradient correction in the case of widely used one-hot encoded genomic sequence data where the simplex is a 3D plane within a 4D space, for each nucleotide.

Given $\overrightarrow{n} = \frac{1}{\sqrt{4}}(\hat{i} + \hat{j} + \hat{k} + \hat{l})$ is a normal vector to the simplex plane (Eq. 1) and $\overrightarrow{G}$ is the gradient of function $f$,

$$\overrightarrow{G} = \frac{\partial f}{\partial x}\,\hat{i} + \frac{\partial f}{\partial y}\,\hat{j} + \frac{\partial f}{\partial z}\,\hat{k} + \frac{\partial f}{\partial w}\,\hat{l}, \tag{2}$$

we can correct $\overrightarrow{G}$ by removing the unreliable orthogonal component, according to:

$$\vec{G}_{\text{corrected}} = \vec{G}_{\parallel} = \vec{G} - \vec{G}_{\perp} = \vec{G} - (\vec{G} \cdot \vec{n})\,\vec{n}$$
$$= (\frac{\partial f}{\partial x} - \mu)\hat{i} + \dots + (\frac{\partial f}{\partial w} - \mu)\hat{l} \tag{3}$$

where $\mu = \frac{1}{A}\sum_i \frac{\partial f}{\partial x_i}$ and $A$ is the dimensionality of the one-hot categories. For DNA, $A = 4$. For proteins, $A = 20$. Hence, our proposed gradient correction—subtracting the original gradient components by the mean gradients across components—is general for all data with categorical inputs.

To implement the gradient correction for an attribution map that has a shape $(N, L, A)$, where $N$ is the number of attribution maps, a correction using NumPy [27] can be achieved with: \$ attr_map = attr_map - np.mean(attr_map, axis=2, keepdims=True).

### Data

#### *Synthetic data*

The synthetic binary classification data from Ref. [19] reflects a simple billboard model of gene regulation. Briefly, 20,000 synthetic sequences, each 200 nucleotides (nt) long, were embedded with known motifs in specific combinations in an equiprobable sequence model. Positive class sequences were generated by sampling a sequence model embedded with 3 to 5 "core motifs," randomly selected with replacement from a pool of 10 position frequency matrices, which include the forward and reverse-complement motifs for CEBPB, Gabpa, MAX, SP1, and YY1 proteins from the JASPAR database [20]. Negative class sequences were generated following the same steps with the exception that the pool of motifs include 100 non-overlapping "background motifs" from the JASPAR database. Background sequences can thus contain core motifs; however, it is unlikely to randomly draw motif combinations that resemble a positive regulatory code. The dataset is randomly split into training, validation and test sets with a 0.7, 0.1, and 0.2 split, respectively. The machine learning task is to predict class membership of one-hot sequences 200 nt in length.

#### *ChIP-seq data*

Transcription factor (TF) chromatin immunoprecipation sequencing (ChIP-seq [28]) data was processed and framed as a binary classification task. Similar to the synthetic dataset, the input is 200 nt DNA sequences and the output is a single binary prediction of TF binding activity. Positive-label sequences represent the presence of a ChIP-seq peak and negative-label sequences represent peaks for non-overlapping DNase I hypersensitive sites from the same cell type that do not overlap with any ChIP-seq peaks. Ten representative TF ChIP-seq experiments in a GM12878 cell line and a DNase-seq experiment for the same cell line were downloaded from ENCODE [29], for details see Additional file 1: Table S1. BEDTools [30] was used to identify non-overlapping DNase-seq peaks and the number of negative sequences were randomly down-sampled to exactly match the number of positive sequences, keeping the classes balanced. The dataset was split randomly into training, validation, and test set according to the fraction 0.7, 0.1, and 0.2, respectively.

### Models

For the analysis of synthetic data and ChIP-seq data, we used two different base CNN architectures, namely CNN-shallow and CNN-deep, each with two variations—rectified linear units (ReLU) or exponential activations for the first convolutional layer, while ReLU activations are used for other layers—resulting in 4 models in total. CNN-shallow is a network that is designed with an inductive bias to learn interpretable motifs in first layer filters with ReLU activations [31]; while CNN-deep has been empirically shown to learn distributed motif representations. Both networks learn robust motif representations in first layer filters when employing exponential activations [19].

All models take as input one-hot-encoded sequences (200 nucleotides) and have a fully-connected output layer with a single sigmoid output for this binary prediction task. The hidden layers for each model are:

1. CNN-shallow

    1. Convolution (24 filters, size 19, stride 1, activation)

       Max-pooling (size 50, stride 50)
    2. Convolution (48 filters, size 3, stride 1, ReLU)
       Max-pooling (size 2, stride 2)
    3. Fully-connected layer (96 units, stride 1, ReLU)

2. CNN-deep

    1. Convolution (24 filters, size 19, stride 1, activation)
    2. Convolution (32 filters, size 7, stride 1, ReLU)
       Max-pooling (size 4, stride 4)
    3. Convolution (48 filters, size 7, stride 1, ReLU)
       Max-pooling (size 4, stride 4)
    4. Convolution (64 filters, size 3, stride 1, ReLU)
       Max-pooling (size 3, stride 3)
    5. Fully-connected layer (96 units, stride 1, ReLU)

We incorporate batch normalization [32] in each hidden layer prior to activations; dropout [33] with probabilities corresponding to CNN-shallow (layer1 0.1, layer2 0.2) and CNN-deep (layer1 0.1, layer2 0.2, layer3 0.3, layer4 0.4, layer5 0.5); and $L2$-regularization on all parameters of hidden layers (except batch norm) with a strength equal to $1e-6$.

We uniformly trained each model by minimizing the binary cross-entropy loss function with mini-batch stochastic gradient descent (100 sequences) for 100 epochs with Adam updates using default parameters [34]. The learning rate was initialized to 0.001 and was decayed by a factor of 0.2 when the validation area under the curve (AUC) of the receiver-operating characteristic curve did not improve for 3 epochs. All reported performance metrics are drawn from the test set using the model parameters from the epoch which yielded the highest AUC on the validation set. Each model was trained

50 times with different random initializations according to Ref. [35]. All models were trained using a single P100 GPU; each epoch takes less than 2 seconds.

### Evaluating attribution methods

#### *Attribution methods*

To test the efficacy of attribution-based interpretations of the trained models, we generated attribution scores by employing saliency maps [8], integrated gradients [9], SmoothGrad [10], and expected gradients [11]. Saliency maps were calculated by computing the gradient of the predictions with respect to the inputs. Integrated gradients were calculated by integrating the saliency maps generated from 20 linear interpolation points between a null reference sequence (i.e., all zeros) and a query sequence. Smooth-Grad was employed by averaging the saliency maps of 25 variations of a query sequence, which were generated by adding Gaussian noise (zero-centered with a standard deviation of 0.1) to all nucleotides—sampling and averaging gradients for data that lives off of the simplex. For expected gradients, we averaged the integrated gradients across 10 different reference sequences, generated from random shuffles of the query sequence. Attribution maps were visualized as sequence logos with Logomaker [36].

#### *Quantifying interpretability on synthetic data*

Since synthetic data contains ground truth of embedded motif locations in each sequence, we can directly test the efficacy of the attribution scores. We calculated the similarity of the attribution scores with ground truth using 3 metrics: cosine similarity, area under the receiver-operating characteristic curve (AUROC) and the area under the precision-recall curve (AUPR). Cosine similarity uses a normalized dot product between vector of positions in a given attribution map and the corresponding ground truth vector; the more similar the two maps are, the closer their cosine similarity is to 1. This is done on a per sequence basis. We subtract 0.25 from the ground truth probability matrix to "zero out" non-informative positions and obtain ground truth "importance scores." Thus, cosine similarity focuses on the positions where ground truth motifs are embedded. Interpretability AUROC and AUPR were calculated according to [19], by comparing the distribution of attribution scores in nucleotides belonging to motifs (positive class) and those not associated with any ground truth motifs (negative class). Briefly, we first multiply the attribution scores ($S_{ij}$) and the input sequence ($X_{ij}$) and reduce the dimensions to get a single score per position, according to $C_i = \sum_j S_{ij} X_{ij}$, where $j$ is the alphabet and $i$ is the position, a so-called grad-times-input. We then calculate the information of the ground truth probabilities $M_{ij}$ at each position, according to $I_i = \log_2 4 - \sum_j M_{ij} \log_2 M_{ij}$. Positions that are given a positive label are defined by $I_i > 0.1$ (i.e., 5% of maximum information content for DNA), while positions with an information content of zero are given a negative label. The AUROC and AUPR is then calculated for each sequence using the distribution of $C_i$ at positive label positions against negative label positions.

Each metric captures different aspects of the quality of the attribution maps. For instance, cosine similarity focuses on true positive positions and uses the full gradient vector associated with each sequence. On the other hand, AUROC and AUPR use a single component of the gradient, i.e., the observed nucleotide, due to the grad-times-input.

AUROC and AUPR also focus on a different balance between true positives with either false positives or recall, respectively. Unlike in computer vision, where important features are hierarchical (i.e., edges, textures, and shapes) and extend across several correlated pixels, synthetic genomics data allows us to quantitatively assess the efficacy of attribution maps with "pixel-level" ground truth.

### Quantifying interpretability on ChIP-seq data

For ChIP-seq data, quantitative analysis of interpretability performance is challenging due to a lack of ground truth. We circumvent this by developing a plausible proxy that could serve as ground truth, i.e., ensemble-averaged saliency maps. For each base CNN model, we trained an ensemble of 50 models—each with a slightly different architecture and different random initializations. We achieved slight variations in the architecture by using a different numbers of convolutional filters in the first layer: we trained five models for each of the following ten choices for the number of filters: [12, 14, 16, 18, 20, 22, 24, 26, 28, 30]. Additional variation was coming from initial weights that were randomly initialized according to Ref. [35]. After training and calculating saliency maps for each of these individual models, we then averaged the saliency maps across all 50 models.

For each position $i$ in each sequence, we treated the saliency scores from an individual model as a vector $\vec{S_i}$ with 4 components. The ensemble-average saliency vector $\vec{S_i^{ens}}$ of the same dimension is used to calculate the difference: $\vec{\Delta_S} = \vec{S_i} - \vec{S_i^{ens}}$. We then calculate the L2-norm of $\vec{\Delta_S}$, i.e., $||\Delta_S||_{L_2}$. This score essentially captures how different a saliency map is to the ground truth proxy at the $i$th position in a sequence. To quantify the improvement in saliency maps after the gradient correction, we calculate the percent decrease of $||\Delta_S||_{L_2}$ before and after the correction, according to: $1 - ||\Delta_S||_{L_2}{}^{after}/||\Delta_S||_{L_2}{}^{before}$. We call this the *ensemble difference reduction*.

### Calculating gradient angles

The sine of the angle between a gradient vector and the simplex plane is given by $sin(\alpha) = G_\perp/||G||_{L_2}$, where $||G||_{L_2}$ is the L2-norm of the vector $\vec{G}$, and $G_\perp$ is the orthogonal component of the same vector with respect to the simplex plane. Component $G_\perp$ can be calculated according to: $\vec{G_\perp} = \vec{G} \bullet \vec{n}$, where $\vec{G}$ is given in Eq. 2 and the normal vector for the simplex plane is given by $\vec{n} = \frac{1}{2}(\hat{i} + \hat{j} + \hat{k} + \hat{l})$.

### Additional analysis

#### DeepSTARR—enhancer function with STARR-seq

We acquired the DeepSTARR dataset from Ref. [5]. This consists of a multi-task regression of enhancer activity for STARR-seq [37] data, with 2 tasks that correspond to developmental enhancers (Dev) and housekeeping enhancers (HK). We replicated the DeepSTARR model and trained it on this dataset, which consists of 402,296 training sequences each 249 base-pairs long. Adam optimizer was used with a learning rate of 0.002, and we employed early stopping with a patience of 10 epochs and a learning rate decay that decreased the learning rate by a factor of 0.2 when the validation loss did not improve for 3 epochs. We recovered similar performance, i.e., Pearson's $r$ of 0.68 and 0.75 and a Spearman rho of 0.65 and 0.57 for tasks Dev and HK, respectively. These values are close to the published values of the original DeepSTARR model. We also trained

a modified DeepSTARR where the first layer filter activations were set with exponential activations (DeepSTARR-exp). Training was less stable with the default DeepSTARR settings, so we lowered the learning rate to 0.0003 and added a small dropout of 0.1 after the max-pooling layers in each convolutional block. DeepSTARR-exp achieved a comparable test performance of Pearson's *r* of 0.68 and 0.76 and a Spearman rho of 0.66 and 0.58 for tasks Dev and HK, respectively. For each model, saliency maps were generated for all test sequences by calculating the derivative of the prediction for a respective class with respect to the inputs. These saliency maps were used to generate the angle histogram plot (Additional file 1: Fig. S8a) as well as sequence logos (Additional file 1: Fig. S8b-e). We sub-selected sequence logos based on sequences that contained high angles and demonstrated a compelling visualization of motifs (with low spurious noise) upon gradient correction.

### Basset—chromatin accessibility classification with DNase-seq

We acquired the Basset dataset from Ref. [22]. This consists of a multi-task classification of chromatin accessibility sites across 161 cell types/tissues measured experimentally via DNase-seq [38]. We acquired trained weights for a Basset model trained with ReLU activations and exponential activations in first layer filters in Ref. [19]. For each model, saliency maps were generated for the first 25,000 test sequences by calculating the derivative of the prediction for the highest predicted class with respect to the inputs. These were used to generate the angle histogram plot (Additional file 1: Fig. S9a) as well as sequence logos (Additional file 1: Fig. S9b-f). We sub-selected sequence logos based on sequences that contained high angles and demonstrated a compelling visualization of motifs (with low spurious noise) upon gradient correction.

### GOPHER—chromatin accessibility profile prediction with ATAC-seq

We acquired the test data and the trained CNN-base and CNN-32 models with exponential activations and ReLU activations from Ref. [23]; a total of 4 models. Each CNN takes as input 2kb length sequences and outputs a prediction of normalized read-coverage for 15 ATAC-seq bigWig tracks (i.e., log-fold over control). We calculated gradients of the mean predictions for the PC-3 cell line for sequences that are centered on an IDR peak called by ENCODE data processing pipeline [29]. These saliency maps were used to generate the angle histogram plot (Additional file 1: Fig. S10a) as well as sequence logos (Additional file 1: Fig. S10b-e). We sub-selected sequence logos based on sequences that contained high angles and demonstrated a compelling visualization of motifs (with low spurious noise) upon gradient correction.

## Supplementary Information

---

**Additional file 1.** Supplementary Tables S1, Figures S1-S13, and Notes S1 and S2.

**Additional file 2.** Review history.

---

**Acknowledgements**

This work was supported in part by funding from the NIH grant R01HG012131 and the Simons Center for Quantitative Biology at Cold Spring Harbor Laboratory. This work was performed with assistance from the US National Institutes of Health Grant S10OD028632-01. The authors would like to thank Ziqi (Amber) Tang and Shushan Toneyan for help

generating saliency maps for the CNN models trained on ATAC-seq data. The authors would also like to thank Justin Kinney, David McCandlish, Anna Posfai, and members of the Koo lab for helpful discussions.

**Peer review information**

**Review history**
The review history is available as Additional file 2.

**Authors' contributions**
AM discovered the gradient correction. AM and PKK conceived of the experiments. AM and CR conducted the experiments and analyzed the results. All authors interpreted the results and wrote the manuscript. The authors read and approved the final manuscript.

**Availability of data and materials**
The source code to reproduce analyses in this paper is available under the MIT license at GitHub [39] (https://github.com/p-koo/GradientCorrection). Model weights and processed data, including DeepSTARR [5], Basset [22] and ChIP-seq analysis, are available at Zenodo [40] (https://doi.org/10.5281/zenodo.7011631).

## Declarations

**Ethics approval and consent to participate**
Not applicable.

**Competing interests**
The authors declare that they have no competing interests.

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

## 

