## [**Additional file 2.** Review history. · Genome Biology]

Review History

First round of review

Reviewer 1

Were you able to assess all statistics in the manuscript, including the appropriateness of statistical tests used? There are no statistics in the manuscript.

Comments to author:

Here, the authors present a simple but clever method for correcting attribution-based interpretability methods for deep learning models on biological sequences. Their method comes from the observation that genomic and proteomic data are usually one-hot categorical variables, but neural networks have the capability to learn functions outside of this data simplex. The authors hypothesize that deep learning architectures might be noisy off this data simplex, and suggest that by taking a probabilistic interpretation of inputs (i.e. each position must sum up to 1), this noise can be removed simply by subtracting the gradient of a pure uniform expectation across nucleotides/residues.

Overall, this is an interesting and intuitive proposal, and it is particularly attractive because of its ease of implementation - especially as these interpretation methods are very popular in computational biology, a simple correction like this has utility to the field. The data presented in the paper follows a convincing sequence in supporting this claim - the authors establish that off-simplex behavior is an issue in models trained on both synthetic and real data, show quantitative improvements in importance score in synthetic data where the ground truth is known, and show convincing qualitative results on models trained on real data. The work does invite some open questions however - first, I'm curious how alternative schemes that actually expose these neural networks to probabilistic inputs (e.g. mix-up augmentation during training) would compete here, and second, I'm not fully convinced that the probabilistic interpretation of these categorical inputs fully describes off-simplex behavior (e.g. the neural network's function doesn't have to be restricted to inputs that sum up to 1, and it might have different aberrant behaviors for inputs that violate this assumption). However, since this is just a short report, I don't think it's necessary for the authors to address these additional questions.

Reviewer 2

Were you able to assess all statistics in the manuscript, including the appropriateness of statistical tests used? Yes, and I have assessed the statistics in my report.

Comments to author:

The manuscript by Majdandzic, Rajesh, and Koo describes method to correct gradient-based interpretations of deep neural networks in genomics research. This work identified an important issue with gradient-based attribution methods for DNNs applied in genome sequence data where the input data are categorical variables. Specially, the authors showed those gradient-based methods introduce a random gradient component orthogonal to the simplex input when applied

to DNNs that have off-simplex representation abilities. The authors introduced a statistical correction to effectively remove the off-simplex gradient noise. They demonstrated the utility of the method on interpreting several convolutional neural networks (CNN) that were trained on both synthetic and real ChIP-seq data. They showed the method yield improved attribution maps and such improvement is proportional to the extent of off-simplex random noise, which could be quantified by the angles between the gradient and the simplex surface.

Comments:

1. In the context of genomics, silico mutagenesis is an alternative approach to interpret deep learning model. In this case, the problem of gradient-based attribution does not apply. It would be good to discuss this point.
2. Can the authors further demonstrate how the extent of off-simplex random noise differs across genome in the real datasets and why it happens during training?
3. In supplementary figure 11. The authors showed that initialization is important for DNNs on synthetic data. Can the authors demonstrate that on real data?

GBIO-D-22-01613

Dear Dr. Kevin Pang,

We thank the reviewers for their insightful comments and suggestions. Based on their suggestions, we have added clarifying text and a new analysis to examine the influence of initialization on the magnitude of off-simplex gradient angles for DNNs applied to real data. Please refer to our detailed point-by-point response below (reviewer comments are given in blue and our responses are in black).

Sincerely,

Peter K. Koo
Assistant Professor
Simons Center for Quantitative Biology
Cold Spring Harbor Laboratory

Response to Reviewer comments:

Reviewer #1: Here, the authors present a simple but clever method for correcting attribution-based interpretability methods for deep learning models on biological sequences. Their method comes from the observation that genomic and proteomic data are usually one-hot categorical variables, but neural networks have the capability to learn functions outside of this data simplex. The authors hypothesize that deep learning architectures might be noisy off this data simplex, and suggest that by taking a probabilistic interpretation of inputs (i.e. each position must sum up to 1), this noise can be removed simply by subtracting the gradient of a pure uniform expectation across nucleotides/residues.

Overall, this is an interesting and intuitive proposal, and it is particularly attractive because of its ease of implementation - especially as these interpretation methods are very popular in computational biology, a simple correction like this has utility to the field. The data presented in the paper follows a convincing sequence in supporting this claim - the authors establish that off-simplex behavior is an issue in models trained on both synthetic and real data, show quantitative improvements in importance score in synthetic data where the ground truth is known, and show convincing qualitative results on models trained on real data. The work does invite some open questions however -

"first, I'm curious how alternative schemes that actually expose these neural networks to probabilistic inputs (e.g. mix-up augmentation during training) would compete here and second, I'm not fully convinced that the probabilistic interpretation of these categorical inputs fully describes off-simplex behavior (e.g. the neural network's function doesn't have to be restricted to inputs that sum up to 1, and it might have different aberrant behaviors for inputs that violate this assumption). However, since this is just a short report, I don't think it's necessary for the authors to address these additional questions

These are valid questions. However, in this work, we elected to focus solely on one-hot genomics data, which resides on the vertices of a probabilistic simplex. As mentioned by the reviewer, mixup is a way to sample other areas along the simplex and there are other strategies to probe off-simplex function behavior; this requires perturbations that push the inputs off of the simplex, such as adversarial noise, Gaussian noise, etc. In previous preliminary work, we have explored the effectiveness of such regularization strategies – such as mixup, manifold mixup, and randomized smoothing, among others – in enhancing the efficacy of attribution analysis over standard training [1,2]. These regularization methods can help the model learn a flatter function around the one-hot data, which could potentially mitigate off-gradient noise and lead to smoother functions overall. While the extent of off-simplex gradient angles was not examined in this preliminary study, we plan to incorporate it into our expanded analysis.

In our revised manuscript, we have added a brief statement acknowledging the potential benefit of incorporating these training strategies to reduce off-simplex gradient noise.

"In future research, we plan to explore training strategies that can directly address the off-simplex function behavior of DNNs, such as `\textit{mixup}` `\cite{zhang2017mixup}`, `\textit{manifold mixup}` `\cite{verma2019manifold}`, and `\textit{randomized smoothing}` `\cite{cohen2019certified}`."

[1] Labelson, Ethan L., Rohit Tripathy, and Peter K. Koo. "Towards trustworthy explanations with gradient-based attribution methods." NeurIPS 2021 AI for Science Workshop. 2021.
url: <https://openreview.net/forum?id=LGgo0wPM2MF>

[2] Majdandzic, Antonio, et al. "Selecting deep neural networks that yield consistent attribution-based interpretations for genomics." Machine Learning in Computational Biology. PMLR, 2022.
url: <https://proceedings.mlr.press/v200/majdandzic22a.html>

Reviewer #2: The manuscript by Majdandzic, Rajesh, and Koo describes method to correct gradient-based interpretations of deep neural networks in genomics research. This work identified an important issue with gradient-based attribution methods for DNNs applied in genome sequence data where the input data are categorical variables. Specially, the authors showed those gradient-based methods introduce a random gradient component orthogonal to the simplex input when applied to DNNs that have off-simplex representation abilities. The authors introduced a statistical correction to effectively remove the off-simplex gradient noise. They demonstrated the utility of the method on interpreting several convolutional neural networks (CNN) that were trained on both synthetic and real ChIP-seq data. They showed the method yield improved attribution maps and such improvement is proportional to the extent of off-simplex random noise, which could be quantified by the angles between the gradient and the simplex surface.

Comments:

1. In the context of genomics, silico mutagenesis is an alternative approach to interpret deep learning model. In this case, the problem of gradient-based attribution does not apply. It would be good to discuss this point.

Agreed. In the revised manuscript, we have added clarifying text to highlight that forward propagation methods, such as *in silico* mutagenesis, are not susceptible to off-simplex gradient noise. Specifically, in the Conclusion section, we include:

“On the other hand, attribution methods that are based only on forward propagation, such as *in silico* mutagenesis, do not require these corrections as the DNN’s behavior off the simplex does not affect predictions, because all data, including test data, lives on the simplex.”

2. Can the authors further demonstrate how the extent of off-simplex random noise differs across genome in the real datasets and why it happens during training?

In the revised manuscript, we highlight the extent of off-simplex random noise across the genome via a summary figure that shows a histogram for gradient angles from DNN models trained on ChIP-seq data, Basset models trained on chromatin accessibility data as a binary classification, DeepSTARR model trained on a regression of enhancer activity from STARR-seq data, and a DNN that predicts base-resolution ATAC-seq profiles. In each case, the distribution of angles are broad and anecdotal examples of attribution maps are shown to benefit from our proposed gradient correction.

In the Results section, we have included:

“Strikingly, the extent of off-simplex gradient angles, and hence the occurrence of random attribution noise, was observed consistently across all of the models and datasets that were investigated (Additional file 1: Figs. S7-S10).”

In terms of why it happens during training, we are not exactly sure why it happens. We speculate that it arises because DNNs fit a function everywhere in 4D Euclidean space. The initialization provides random function behavior both along the simplex and off the simplex, but there is no data to guide function behavior off the simplex during training. One way this might be combatted is through data augmentations that provide more examples along the simplex (i.e. mixup) or examples off the simplex (i.e. randomized smoothing). An in-depth study is beyond the scope of this Short Report, but we note that we have begun to explore the impact of these training methods [1]. In the revised manuscript, we provide a brief discussion about this avenue for future research.

In the Conclusion section, we include:

"In future research, we plan to explore training strategies that can directly address the off-simplex function behavior of DNNs, such as *mixup* [Zhang2017mixup], *manifold mixup* [Verma2019manifold], and *randomized smoothing* [Cohen2019certified]."

3. In supplementary figure 11. The authors showed that initialization is important for DNNs on synthetic data. Can the authors demonstrate that on real data?

In the revised manuscript, we have added a similar analysis for how the gradient angles are sensitive to the magnitude of the random initializations using DeepSTARR, which was trained on *in vivo* STARR-seq data (see Additional file 1: Note S1, specifically Fig. S12). This result shows good agreement with what was observed using synthetic data.